# The Effect of Polishing, Glazing, and Aging on Optical Characteristics of Multi-Layered Dental Zirconia with Different Degrees of Translucency

**DOI:** 10.3390/jfb14020068

**Published:** 2023-01-28

**Authors:** Flavia Roxana Toma, Sorin Daniel Porojan, Roxana Diana Vasiliu, Liliana Porojan

**Affiliations:** 1Department of Dental Prostheses Technology (Dental Technology), Center for Advanced Technologies in Dental Prosthodontics, Faculty of Dental Medicine, “Victor Babeș” University of Medicine and Pharmacy Timișoara, Eftimie Murgu Sq. No. 2, 300041 Timisoara, Romania; 2Department of Oral Rehabilitation (Dental Technology), Center for Advanced Technologies in Dental Prosthodontics, Faculty of Dental Medicine, “Victor Babeș” University of Medicine and Pharmacy Timișoara, Eftimie Murgu Sq. No. 2, 300041 Timisoara, Romania

**Keywords:** optical properties, color changes, multilayer translucent zirconia

## Abstract

(1) Background: Considering that the appearance of a dental material is an important factor that contributes to the success of prosthetic restorations. The purpose of this study is to investigate the optical properties and color changes among the layers of three commercial zirconias, to compare the aspect of the polished and glazed surfaces before and after aging and to evaluate the effects of hydrothermal degradation on their aesthetics. (2) Methods: Forty-eight plate-shaped samples were sectioned from presintered blocks of each multilayer translucent zirconia with different Yttrium content: Ceramill Zolid fx ML (5 mol%) = CeZ, STML (4 mol%) = STM, IPS e.maxZirCAD CEREC/in Lab MT Multi (4 mol% + 5 mol%) = IPZ. The samples were sintered according to the recommendation of each manufacturer, and half (24) of them were polished and the other half (24) glazed on one of the surfaces. Each type was equally divided into one control and one aging group, and, for each material, this resulted in four groups (n = 12): polished-control, polished-autoclaved glazed-control, and glazed-autoclaved. The artificial aging was carried out with an autoclave and distilled water at 134 °C, 0.2 MPa for 1 h, and for optical parameters (TP, CR, OP) and color change (ΔE*) measurements on a black and white background in a CIE L*a*b* color system, a spectrophotometer was used. The specimens were evaluated in incisal, cervical, and medium areas on polished and glazed samples before and after the aging stage. Statistical analysis was achieved with a two-way ANOVA test, the unpaired *t*-test, and the paired *t*-test. (3) Results: Before and after aging, the mean TP values for polished samples were higher than the glazed ones. After aging, the mean TP values increased for all groups (except polished CeZ), and significant differences were reported for polished STM, IPZ. After LTD, the opalescence registered an increase for almost all groups (except polished CeZ, polished and glazed IPZ—medium area), and significant differences were reported for almost all groups (except STM—incisal, IPZ—cervical, medium areas). The levels of color change were between extremely slight to perceivable. (4) Conclusions: Optical properties of the selected multilayer zirconia were influenced by polishing and glazing as surface treatment and affected by artificial aging (CeZ the least); perceivable color changes for polished STM, IPZ were detected.

## 1. Introduction

By increasing the Yttrium (Y_2_O_3_) and cubic phase content, which implies an increase in translucency and microstructure modification, the generation of two translucent zirconia is introduced: 5Y-TZP (super-high translucent)—partially stabilized zirconia with 5 mol% Y_2_O_3_ and a more than 50% cubic phase and 4Y-TZP (super-translucent)—partially stabilized zirconia with 4 mol% and a more than 25% cubic phase [1,2]. They are available as presintered discs or blocks, which are pre-shaded in different chromatic gradients [3]. For 5Y-TZP, the amount of tetragonal phase is decreased, leading to a higher translucency and a lower ability of tetragonal (t) to monoclinic (m) phase transformation, with a reduction in toughness. The continuous development of these aesthetic materials aims to reproduce the color and translucency of natural teeth as accurately as possible and maintain these optical properties throughout their functional lifetime in the oral environment, both being a challenge. Additionally, the different treatments and methods of processing and finishing may influence color stability over time and limit the quality and longevity of dental restorations [4]; for this, further studies on multi-layered zirconia are needed.

Cubic crystals are larger than tetragonal ones, with fewer grain boundaries and residual porosities, having an isotropic arrangement that allows incident light to pass more evenly through the material in all spatial directions, leading to higher translucency [5]. Tetragonal zirconia has an anisotropic crystal structure (induce birefringence)—light passes through the material only in certain directions. When it reaches a grain boundary, it is partially or totally scattered, absorbed, and diffusely reflected, causing a reduction in transmitted light and translucency. The 3Y-TZP zirconia is more opaque than 4Y-/5Y-TZP [6,7]. Summarizing the reports of several studies on different types of zirconia, the super-translucent one presents particles with a size of 0.5–1 µm and a cubic phase content of 25–53 wt%, and the super-high translucent zirconia may contain grains larger than 1 µm, differently distributed, and a 60–71 wt% cubic phase, making them highly transparent but more susceptible to fracture [6,8]. The translucency and opacity depend on the amount and the way the incident light is transmitted, reflected, refracted, scattered, and absorbed. If the light is scattered and most of it is diffusely transmitted through the material, it will have a translucent appearance; in the contrary situation, most of it is absorbed and diffusely reflected, and the appearance of the zirconia will be opaque [9,10].

The color parameters and appearance of dental zirconia may be influenced by extrinsic factors—food and drink colorants (caffeine, tannins, nicotine) and by intrinsic factors—the microstructure (Yttrium and phase content, grain size, density, the amount and type of pigment), surface treatment, aging behavior, water absorption, and surface roughness [11].

The multilayer zirconia is composed of polychromatic layers: cervical (dentin), medium (transition), and incisal (enamel); the dentin layer looks more opaque due to the lowest light permeability, and the enamel layer appears the most transparent—mimicking as much as possible the chroma, translucency, or opacity, the opalescence of the natural teeth.

Depending on the Yttrium amount and chemical composition, the layers occur as different formulations; therefore, their physical properties in a material may be different [12,13].

In the case of the first multi-layered zirconia (Katana, Kuraray Noritake), all layers were similar, the difference between them being given only by the color of the pigment [14]. Later, materials with varied microstructures among the layers were introduced (IPS e.max ZirCAD CEREC/in Lab MT Multi). There are contradictory opinions regarding the effects of the embedding of oxide pigments on the tetragonal particle stability and optical properties of zirconia, as they have a different refractive index and cause a decrease in translucency [15]. Alumina-Al_2_O_3_ (0.5–1 wt%) influences the microstructure, enhances the densification rate, delays the hydrothermal degradation, and decreases the translucency due to its segregation on grain boundaries [16,17]. Hafnium-Hf O_2_ (≤5 wt%), other oxides (≤1 wt%)—Na_2_O, SiO, Er_2_O_3_, Fe_2_O_3_, and transition and rare earth metals may be contained in the commercial zirconias [18,19,20].

Glazing and polishing as surface treatments not only improve aesthetics but also reduce the wear of the opposite teeth, bacterial adhesion, and inflammatory tissue reactions [21,22]. The glazing layer minimizes abrasiveness by sealing pores and smoothing the surface, preserving color stability by protecting the material from staining [23]. After aging, it can be affected directly or due to the alteration of the underlying zirconia, with cracks, thinning, or detachment from the surface; it depends on the microstructure, surface smoothness, and interface conditions, with a great influence on external appearance and optical properties.

Previous studies have reported that the polishing ability is influenced by microstructure [24]. Slight polishing, as a result of residual stress, may cause t–m transformation zones around scratches, with an increase in the monoclinic phase and roughness [25] and a decrease in translucency. Fine grinding (15–30 µm) induces no evident defects [26,27].

It is essential to evaluate the long-term stability of zirconia in an intraoral environment, and for this, steam autoclaves at increased temperatures (120–140 °C) are widely used to achieve the effective aging of the material [28]. Several in vitro studies have evaluated this procedure, which presents some limitations; therefore, the results should be extrapolated with caution in clinical conditions [29]. The aging procedure (low-temperature degradation (LTD)) is accompanied by a slow t-to-m transformation, micro-cracks, grain pull-out, and deterioration of the surface exposed to the humid environment [30,31]. It was observed that zirconia with higher cubic content underwent almost no transformation, and it was least affected by hydrothermal degradation [32].

The purpose of the study is to evaluate the optical properties—the dependent variables TP, CR, OP, and color stability (ΔE)—among layers with different microstructures of three types of monolithic multi-layered translucent zirconia related to surface treatment (glazing/polishing) and low hydrothermal degradation. The null hypotheses established for this study are: (a) the optical properties of the materials are not influenced by the surface treatments—before or after aging; (b) the translucency and color stability of the zirconia are affected by the artificial aging procedure.

## 2. Materials and Methods

### 2.1. Specimen Preparation

Three multi-layered dental zirconia with different Yttrium content and translucency were selected for the experimental analyses: Ceramill Zolid fx ML (Amman Girrbach, AG, Koblach, Austria); STML (Katana, Kuraray Noritake Dental, Tokyo, Japan); IPS e.maxZirCAD CEREC/in Lab MT Multi (IvoclarVivadent AG, Schaan, Liechtenstein), marked with the following abbreviations: CeZ, STM, and IPZ, respectively. The properties of the materials are listed in Table 1.

Forty-eight specimens of each material were sectioned from presintered blocks, polished on both sides with abrasive paper (1000–2000 grit), and sintered in a ceramic furnace (LHT 03/17D; Nabertherm, Lilienthal, Germany) at a temperature of 1450–1550 °C, according to the specific recommendation of each manufacturer. The samples were finished on one surface: half of them by glazing (24) and the other half (24) by polishing. Then, each type was divided into a control group and one aging group, obtained after the autoclaving procedure. Four sample groups resulted for each zirconia (n = 12): polished-control, glazed-control, polished-autoclaved, and glazed-autoclaved. The samples were performed by a specialized dental technician, and the treated surfaces (polished/glazed) were evaluated before and after aging in the incisal, medium, and cervical areas by a single operator.

The studied groups are described in a flowchart—Figure 1:

Polishing was performed with a diamond paste (Zirkonzahn Polishing Paste; Zirkonzahn, Norcross, GA, USA), a polishing brush, and a low-speed handpiece. The glazing procedure was achieved in a furnace (Programat P310 G2; IvoclarVivadent, Schaan, Liechtenstein) with IPS Ivocolor Glaze Paste (IvoclarVivadent, Schaan, Liechtenstein). The final dimensions of the specimens, corresponding to the size of the blocks, were 12 × 14 × 1 mm, checked with a digital caliper (Figure 2).

### 2.2. Hydrothermal Aging Protocol

An autoclave (Sterilclave 24 B, Cominox, Carate Brianza, Italy) and distilled water (134 °C, 0.2 MPa) for 1 h was used for the artificial aging procedure. It was found that the 1 h autoclave treatment in these circumstances would have the same impact on zirconia as 3–4 years in the oral environment [36].

### 2.3. Optical and Color Change Measurements

The optical parameters—translucency (TP), contrast ratio (CR), and opalescence (OP)—were recorded under D65 standard illumination on polished and glazed surfaces before and after aging using a spectrophotometer (Easyshade IV; Vita Zahnfabrik, Bad Säckingen, Germany); it is a clinical device that only works in “tooth mode” [37]. The instrument was calibrated before each measurement, and the probe tip was held at 90° to the surface of the sample. The recording was accepted when two identical consecutive readings were obtained on each area (cervical, incisal, medial) in three randomly selected zones of the polished or glazed surfaces. The measurements were performed by a single operator.

To determine the CIE L*a*b* coordinates—the black (b) and white (w) background of the grey card WhiBal G7 (White Balance Pocket Card) was used.

L* is the lightness coordinate (L* = 0 perfect black, L* = 100 perfect white); a* = is the chromatic coordinate in the red (positive value)/green (negative value) axis, and b* = is the chromatic coordinate in the yellow (positive value)/blue (negative value) axis [38,39,40].

TP values result by calculating the color difference on black and white backgrounds according to the formula:TP = [(L*_b_ − L*_w_)^2^ + (a*_b_ − a*_w_)^2^ + (b*_b_ − b*_w_)^2^]^1/2^(1)

TP values may range from 0 (totally opaque) to 100 (totally transparent).

CR defines the opacity and is calculated over a black and white background according to the formula:CR = Y_b_/Y_w_ = [(L* + 16)/116]^3^ × 100(2)

CR values may range from 0 (totally transparent) to 1 (totally opaque).

OP values estimate the difference in red–green and yellow–blue color coordinates between transmitted and reflected light; the values are obtained using the formula:OP = [(a*_b_ − a*_w_)^2^ +(b*_b_ − b*_w_)^2^]^1/2^(3)

ΔE*—The total color change value, signifying the color difference between two stages, was achieved using the formula:ΔE* = [(ΔL*)^2^ + (Δa*)^2^ + (Δb*)^2^]^1/2^
(4)

The recordings were performed for each group in three areas (cervical, medium, incisal).

To report the color change to a clinical standard, ΔE* was converted to NBS units (NBS—The National Bureau of Standards) according to the formula: NBS = ΔE* × 0.92 [34,35,36,37]. In conformity with NBS, the levels of color changes (expressed in NBS units) are: extremely slight change (0.0–0.5), slight change (0.5–1.5), perceivable (1.5–3.0), marked change (3.0–6.0), extremely marked change (6.0–12.0), and change to another color (12.0–more).

### 2.4. Statistical Analysis

Two programs were used—JASP (v.16.2, University of Amsterdam, Amsterdam, The Netherlands) and IBM SPSS Statistics software (IBM, New York, NY, USA). Using descriptive statistics in the first part, the dispersion parameters and central tendency were calculated. The normal distribution of the dependent was verified with the Shapiro–Wilk test; non-parametrical and parametrical tests were applied for a complete perspective. A two-way ANOVA with the replication/Friedman test was applied for the statistical analysis of more than two dependent groups (the areas of a treated surface); the unpaired Student *t*-test for two different groups (the variables registered on polished and glazed surfaces of a material), and the paired Student *t*-test for two time moment states (to compare the same areas of a material—before and after aging). Pearson correlation and regression analyses were performed to evaluate the interdependence between the variables recorded before and after aging. The significance of the Pearson coefficient (r) was related to [0; 0.2]—“very weak”, [0.2; 0.4]—“weak”, [0.4; 0.6]—“moderate”, [0.6; 0.8]—“strong”, and [0.8–1.0]—“very strong”. The coefficient of determinations (r^2^) indicates the percentage of the total variation of the dependent variable (L*, a*, b*, TP, OP) that is explained by the variation of the independent variable (material, aging time). A significance level of α = 0.05 was set.

## 3. Results

The mean L*, a*, b*, TP, CR, OP values and standard deviation (SD), registered on polished and glazed specimens in cervical, medium, and incisal areas before aging can be found in Table 2:

Concerning polished and glazed tested samples, the lowest mean TP values were recorded for STM, with close values between cervical and medium areas; the highest values were for CeZ and for both materials; the between areas tended to increase from cervical to incisal (c < m < i). For IPZ, the order was as follows: c > m < i; the lowest mean values were registered in the medium area. Among areas of the materials, close values were recorded for STM and IPZ in the medium area.

The lowest mean CR values were recorded for CeZ and the highest for STM; the between areas tended to decrease from cervical to incisal (c > m > i) for both materials in the polished and glazed tested samples. For IPZ, the order was as follows: c < m > i, with the highest value in the medium area.

The lowest mean OP values were measured for CeZ and the highest for STM, with intermediate values for IPZ; the between areas tended to decrease from cervical to incisal (c > m > i) for all three materials in polished and glazed tested samples.

The mean L*, a*, b*, TP, CR, OP values and standard deviation (SD), measured on polished and glazed specimens in cervical, medium, and incisal areas after LTD (low-temperature degradation) by autoclaving, are shown in Table 3.

After aging, regarding polished and glazed tested samples, it was observed that the mean TP values increased in the three areas for all materials (less-incisal) except for the CeZ polished samples, where the values decreased (however, it remained the highest among zirconias), and the scale of the values between areas for a material was maintained as before aging. The lowest TP values were measured for STM, with close values between cervical and medium areas, and the highest for CeZ. For IPZ, the order was as follows: c > m < i, with the lowest values in the medium area. The translucency remained almost as before in the incisal area of the polished specimens. Among areas of the materials, close values were recorded for STM and IPZ in the medium area (Figure 3).

Both before and after the aging stage, a two-way ANOVA statistical test (α = 0.05) was applied to compare the TP values of the three areas (incisal, medium, cervical) individually for material-reported significant differences (*p* < 0.05) for CeZ and IPZ on polished and glazed surfaces.

It was found that before and after the autoclaving procedure, the TP values for the polished samples were higher than those for glazed samples for all areas and materials. To perform statistical analysis between the TP values of the polished and glazed groups in the same areas for a material, the unpaired *t*-test was applied. Significant differences (*p* < 0.05) were reported for all materials in the three areas before the aging stage and for STM (all), IPZ (cervical, medium) after the aging stage. The statistical paired *t*-test, regarding TP for the polished control–autoclaved groups and the glazed control–autoclaved groups in each area, revealed significant differences (increases) in terms of translucency between the specimens before and after autoclaving on the polished surfaces in cervical and medium areas for STM and IPZ. Pearson correlation and regression analyses were performed to evaluate the relationships between the variables (L*, a*, b*) and aging time for the polished surfaces of STM (cervical, medium) and IPZ (cervical, medium) (Table 4).

After aging, concerning the polished and glazed tested samples, it was observed that the mean CR values decreased in the three areas for all materials except for the CeZ polished samples, where the values increased (but remained the lowest among the materials). The lowest CR values were measured for CeZ and the highest for STM among the areas; the scale of values for a material was maintained as before aging (c > m > i). For IPZ, the order was: c < m > i, with the highest value in the medium area.

The mean CR values for all groups, recorded in incisal, cervical, and medium areas, are shown in Figure 4.

Both before and after the aging stage, a two-way ANOVA statistical test (α = 0.05) was performed to compare the CR values of the three areas (medium, cervical, incisal) individually on the polished and glazed surfaces of a material. We report significant differences (*p* < 0.05) for CeZ, IPZ on polished and glazed surfaces.

The unpaired *t*-test was applied for statistical analysis between the CR values of the polished and glazed groups in the same area for a material before and after the aging stage. The CR values for the polished samples are lower than those for the glazed samples for all areas and materials, both before and after aging. Significant differences (*p* < 0.05) were reported in the before-aging stage in the three areas of all materials and in the after-aging stage for STM (all), IPZ (cervical, medium areas). The statistical paired *t*-test reported significant differences in terms of CR among the specimens before and after autoclaving in the cervical and medium areas on the polished surfaces for STM and IPZ. After autoclaving, referring to the polished test samples, the OP values decreased for CeZ and IPZ (medium area) and increased for STM and IPZ (cervical, incisal area); the lowest mean OP values were measured for CeZ (5.071–3.313) and the highest for STM (8.184–5.852), with intermediate values for IPZ (6.405–4.179). Close values between cervical and medium areas for STM and IPZ were found. The between areas tended to decrease from cervical to incisal (c > m > i) for all three materials.

Regarding glazed tested samples, it was observed that the mean OP values increased for all materials in the three areas (except the IPZ medium area). The highest values were measured for STM and the lowest for CeZ; very close values among the cervical and medium areas of CeZ and IPZ were found. The between areas tended to decrease from cervical to incisal (c > m > i) for all three materials (Figure 5).

A two-way ANOVA statistical test (α = 0.05) was performed to compare the OP values of the three areas (incisal, cervical, medium) individually on the polished and glazed surfaces for a material; we report significant differences (*p* < 0.05) for most analyses (except IPZ glazed samples) in the before-aging stage and for CeZ polished samples and STM glazed samples in the after-aging stage. The unpaired *t*-test was performed for statistical analysis between the OP mean values in the same areas of the polished and glazed surfaces of a material. Significant differences (*p* < 0.05) were reported before aging for CeZ (all areas), STM (medium area), and IPZ (medium, incisal areas), and the values of the polished surfaces were higher than those of the glazed ones. In the after-aging stage for CeZ (all areas), STM (cervical, medium area), and IPZ (incisal area), the values of the polished surfaces were higher than those of the glazed ones for STM (cervical, medium areas) and lower for CeZ (all) and IPZ (incisal area). Following the aging process, decreases in values were observed for some polished surfaces and increases in values for some glazed ones; the statistical paired *t*-test reported significant differences in terms of OP among the samples before and after autoclaving on the polished surfaces for CeZ, STM (cervical, medium areas), and IPZ (incisal area) and on the glazed surfaces for CeZ, STM (medium area).

Pearson correlation and regression analyses were performed to evaluate the relationships between variables for the polished surfaces of CeZ (c, m areas), STM (c, m areas), and IPZ (i area) and for the glazed surfaces of CeZ (m area), STM (m area) (Table 5).

According to the NBS system, after autoclaving, the levels of color change were between extremely slight to perceivable (Table 6).

## 4. Discussion

The LTD treatment activates and accelerates the t-to-m phase transformation, associated with the deterioration of surfaces exposed to moisture. In Pereira et al.’s systematic review [29], the authors reported that the m-phase contents were about 0–14% before aging and 11–40% after accelerated aging for 10 h. An increase in translucency, explained by the reduction of light scattering from the boundaries of the cubic phase particles [41], was also observed. However, for super-high translucency zirconia, the presence of the cubic phase induces an insignificant hydrothermal degradation, with a minor increase in translucency; the lack of phase transformation and the coarser microstructure determine lower mechanical properties for this zirconia.

After aging, an increase in TP values was reported for all materials (more for STM and less for CeZ) except CeZ on polished surfaces; however, STM remains the zirconia with the lowest translucency, and CeZ with the highest, among materials for both treated surfaces. Previous studies have revealed that during the aging process, the surface roughness increases more for super translucency zirconia (STM-4 mol%) and less or not at all for super-high translucency zirconia (CeZ-5 mol%). However, it has a rougher surface; the material is affected by the scattering effect, and the brightness decreases as the roughness increases [42,43]. An explanation for the decrease in TP and OP values for polished CeZ would be that the high roughness of the material negatively affects the appearance of the surface. For the polished IPZ in the incisal area, the values were almost similar to before, which indicates that this area shows an aging behavior similar to CeZ, having the same microstructure.

In this study, it can also be observed that the cervical (dentin) area is more susceptible to hydrothermal degradation and color changes than the incisal (enamel), probably due to the existence and different behavior of color pigments related to aging [44].

It was found that the increased rate in the polished specimens was higher than in the glazed ones, similarly in the cervical areas compared to the incisal ones.

It seems that the adhesion and sealing capacity of the glaze layer is better on the surface of zirconia with smaller grains than for the materials with higher Yttrium content and larger grains. STM contains smaller particles than CeZ, and perhaps, the glaze layer adheres better and has a more protective role, even if the surface roughness has increased. Nevertheless, this material is most affected by aging, accompanied by an increase in translucency (more evident on polished surfaces).

Following the aging process, notable changes in the color parameters were registered. By the significant increase in TP, a strong positive correlation between variables (b* and aging time) was found for the polished IPZ cervical area—the samples displayed a darker and more yellow appearance. For polished STM, moderate correlations were found—the samples had a redder (in the cervical area) and more yellow (in the medium area) appearance. By the significant increase in OP, a strong correlation between the variables was found for the glazed STM medium area, with more yellow surfaces. By the significant decrease in OP values, for polished CeZ, a moderate correlation was found—the specimens had a more bluish (in the cervical area) and redder (in the medium area) appearance.

At a thickness of 1 mm, it was related that the mean TP values of conventional zirconia range from 4.83–9.10 [45]; For bovine teeth, the values range from 14.7–15.2, and for human teeth, from 15–19 [46,47]. In the current study, the recorded interval was 13.08–14.58.

Referring to CR, it is known that the source of zirconia opacity is alumina-Al_2_O_3_. The perceptible threshold for the human eye ranges from 0.06–0.08 [48]. In the current study, the difference between the mean CR values recorded for CeZ and STM in the cervical and incisal areas on the polished samples was 0.03 (both areas) before aging and 0.03 (cervical), 0.01(incisal) after aging; on the glazed samples, the values were 0.04 (both areas) before aging and 0.03 (cervical), 0.02 (incisal) after aging. According to this information, no dissimilarities are clinically detectable in terms of the opacity of the same areas of these two zirconias.

Opalescence is an optical parameter that denotes the blueness of the reflected light. It was found that restoration is optically optimal if it has an opalescence similar to that of the adjacent tooth [23,48,49]. In the present study, the highest mean OP values were recorded for STM and the lowest for CeZ on glazed and polished samples before and after aging. After the autoclaving procedure, the OP values increased for almost all materials (except polished CeZ, polished and glazed IPZ (medium area)). Significant differences were found among the samples before and after autoclaving for most groups except STM (incisal area) and IPZ (cervical, medium areas) in the polished and glazed surfaces. The mean OP values of tetragonal zirconia were reported in an interval of 1.25–2.83 [49], and for human teeth, 4.8–7.4 [46]. In the current study, the OP mean values ranged from 3.5–7.5 before aging and 3.3–8.1 after aging. It was found that the glaze layer reduces opalescence without affecting translucency [42].

In this study, polishing and glazing as surface treatments influenced the optical properties of the materials before and after aging; thus, the first null hypothesis is rejected.

Based on previous studies, the autoclaving procedure causes a wide range of changes in the mechanical and optical properties of the materials, and for highly translucent zirconia, an opaquer appearance has been reported [50,51,52]. In this study, after aging, the TP values increased for all materials in the glazed and polished samples except for polished CeZ, where the values decreased; therefore, the second null hypothesis—artificial aging affects the translucency and color stability of zirconia—is accepted.

Regarding the aging process, a correlation between the translucency, mechanical properties, and thickness of the material was observed, along with the influence of composition and grain size on the surface treatments [14,32]. According to these findings, future studies are needed to investigate the behavior of the dopants over time. The appearance and changes that occur depend on the thickness of the material. Because zirconia, in particular 4Y-TZP, has an almost similar hardness [53] but lower translucency [54] than lithium disilicate glass–ceramics, a novel alternative such as zirconia-reinforced lithium silicate glass–ceramics can be taken into consideration.

Natural human teeth present a blend of colors and shades, with layers of different thicknesses (dentin, enamel) and varied modes of light transmission. Therefore, the limitations of this study would be: the low number of investigated multi-layered zirconias; only one type of glaze paste was used; the samples had the same uniform thickness; the short exposure time in the autoclave and the effectiveness of the aging method; the lack of certain existing conditions (mechanical/chemical stimuli) in the oral environment.

## 5. Conclusions

Polishing and glazing, as surface treatments, influence the optical properties of zirconia; the translucency of the polished samples was higher than that of the glazed samples before and after aging.After aging, the mean TP and OP values were increased for all materials except polished CeZ; the super-high translucent zirconia was less affected by LTD.The levels of color change were between extremely slight to perceivable.

## Figures and Tables

**Figure 1 jfb-14-00068-f001:**
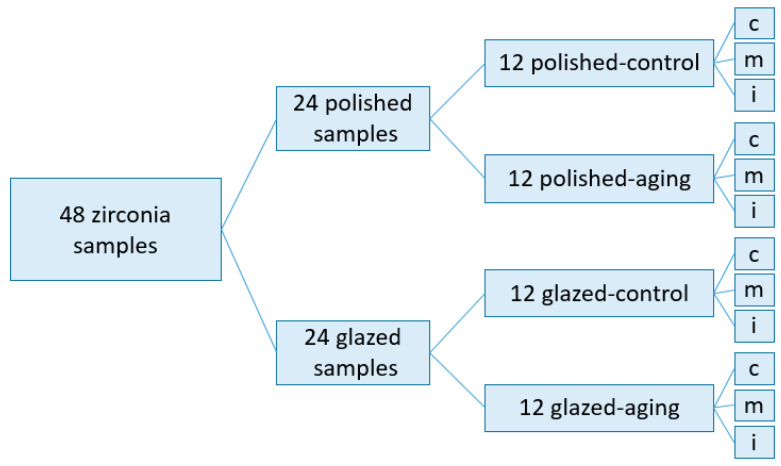
The flowchart in which the distribution of sample groups is described; c = cervical, m = medium, i = incisal.

**Figure 2 jfb-14-00068-f002:**
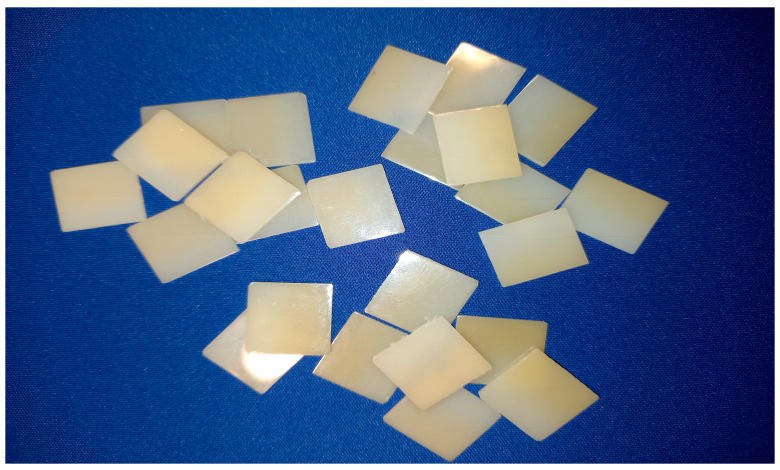
The studied samples of the three types of zirconia.

**Figure 3 jfb-14-00068-f003:**
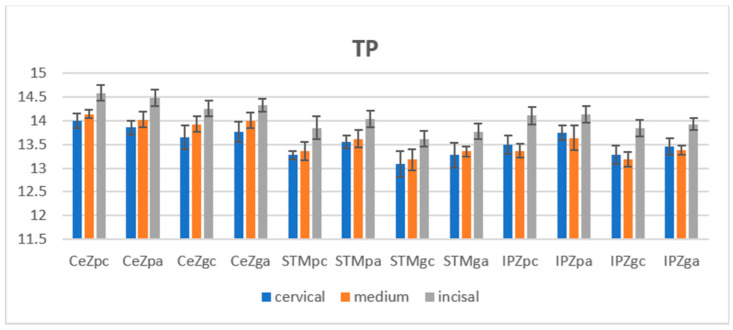
Mean TP values with standard deviation on polished and glazed specimens before and after aging. TP = translucency parameter, CeZ = Ceramill Zolid fx ML, STM = STML, IPZ = IPS e.max ZirCAD CEREC/in Lab MT Multi, pc = polished-control, gc = glazed-control, pa = polished-autoclaved ga = glazed-autoclaved.

**Figure 4 jfb-14-00068-f004:**
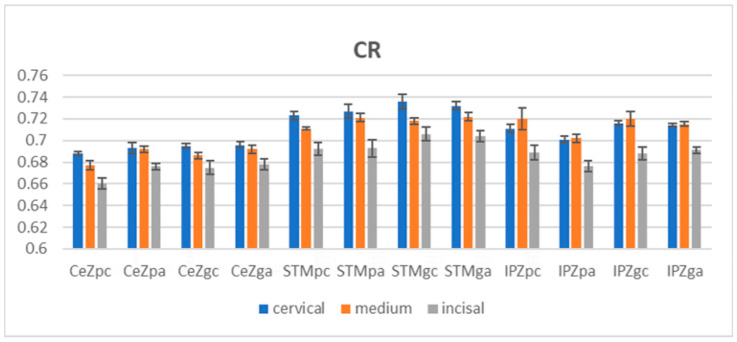
Mean CR values with standard deviation, before and after aging. CR = contrast ratio, CeZ = Ceramill Zolid fx ML, STM = STML, IPZ = IPS e.max ZirCAD CEREC/in Lab MT Multi, pc = polished-control, gc = glazed-control, pa = polished-autoclaved, ga = glazed-autoclaved.

**Figure 5 jfb-14-00068-f005:**
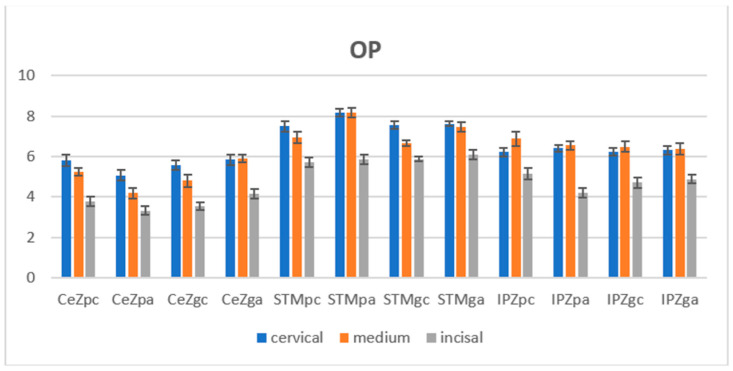
Mean OP values with standard deviation, before and after aging. OP = opalescence parameter, CeZ = Ceramill Zolid fx ML, STM = STML, IPZ = IPS e.max ZirCAD CEREC/in Lab MT Multi, pc = polished-control, gc = glazed-control, pa = polished-autoclaved, ga = glazed-autoclaved.

**Table 1 jfb-14-00068-t001:** The manufacturer’s specification for the studied materials [33,34,35].

Material/Manufacturer	Y_2_O_3_ Content	Other Components	Grain Size	Shade/Translucency
Ceramill Zolid fx ML/ Amman Girrbach, AG, Koblach, Austria (CeZ)	5 mol% (8.5–9.5 wt%)	Hf O_2_ ≤ 5.0 wt% Al_2_O_3_ ≤ 0.5 wt% other oxides ≤ 1.0 wt%	-	Super-high translucency (A2/A3)
STML/Katana, Kuraray Noritake Dental, Tokio, Japan (STM)	4 mol% (7–10.0 wt%)	Hf O_2_ ≤ 5.0 wt% Al_2_O_3_ 0.26 wt% other oxides < 2%	0.2–0.5 µm	Super translucency (A2)
IPS e.max Zir CAD/ Ivoclar Vivadent AG, Schaan, Liechtenstein (IPZ)	4 mol%-cervical (7.10 wt%) 4 + 5 mol%-transition 5 mol%-incisal (8.80 wt%)	Hf O_2_ ≤ 5.0 wt% Al_2_O_3_ ≤ 0.05 wt% other oxides ≤ 1.0 wt%	0.65 µm 0.85 µm	Super-high translucency + Super translucency (A2)

**Table 2 jfb-14-00068-t002:** Mean with SD for CIE L*, a*, b*and TP, CR, OP values—before aging.

Variable	Area	CeZ pc	CeZ gc	STM pc	STM gc	IPZ pc	IPZ gc
L*	c	78.213 ± 0.271	80.813 ± 0.217	76.650 ± 0.288	83.613 ± 0.187	83.288 ± 0.289	83.613 ± 0.278
	m	77.888 ± 0.267	81.700 ± 0.269	78.613 ± 0.289	83.988 ± 0.278	82.100 ± 0.234	83.988 ± 0.348
	i	79.825 ± 0.288	82.825 ± 0.219	80.488 ± 0.219	84.075 ± 0.234	83.463 ± 0.267	84.075 ± 0.234
a*	c	0.013 ± 0.054	−0.075 ± 0.005	−0.738 ± 0.005	0.438 ± 0.005	0.413 ± 0.005	0.438 ± 0.005
	m	−0.850 ± 0.024	−0.938 ± 0.054	−1.363 ± 0.003	0.175 ± 0.109	0.275 ± 0.109	0.175 ± 0.004
	i	−1.113 ± 0.005	−1.025 ± 0.101	−2.388 ± 0.014	−1.413 ± 0.102	−1.613 ± 0.109	−1.413 ± 0.109
b*	c	10.675 ± 0.218	10.413 ± 0.250	26.000 ± 0.207	16.638 ± 0.218	17.125 ± 0.280	16.638 ± 0.267
	m	5.700 ± 0.250	5.850 ± 0.187	20.763 ± 0.267	15.738 ± 0.234	17.825 ± 0.267	15.738 ± 0.207
	i	1.825 ± 0.207	1.938 ± 0.207	12.738 ± 0.280	5.575 ± 0.278	4.538 ± 0.188	5.575 ± 0.256
TP	c	13.996 ± 0.158	13.656 ± 0.25	13.274 ± 0.185	13.083 ± 0.271	13.491 ± 0.191	13.284 ± 0.197
	m	14.139± 0.119	13.929 ± 0.166	13.353 ± 0.16	13.178 ± 0.215	13.366 ± 0.147	13.188 ± 0.16
	i	14.587 ± 0.167	14.258 ± 0.168	13.846 ± 0.24	13.612 ± 0.165	14.108 ± 0.189	13.838 ± 0.175
CR	c	0.688 ± 0.002	0.695 ± 0.002	0.723 ± 0.004	0.736 ± 0.007	0.711 ± 0.004	0.716 ± 0.002
	m	0.677 ± 0.004	0.686 ± 0.003	0.711 ± 0.001	0.718 ± 0.003	0.724 ± 0.01	0.720 ± 0.007
	i	0.66 ± 0.005	0.675 ± 0.006	0.692 ± 0.006	0.706 ± 0.006	0.689 ± 0.007	0.688 ± 0.006
OP	c	5.802 ± 0.192	5.581 ± 0.236	7.486 ± 0.273	7.555 ± 0.203	6.207 ± 0.201	6.239 ± 0.177
	m	5.218 ± 0.18	4.791 ± 0.296	6.929 ± 0.298	6.651 ± 0.143	6.867 ± 0.350	6.486 ± 0.237
	i	3.787 ± 0.217	3.529 ± 0.189	5.704 ± 0.222	5.855 ± 0.116	5.159 ± 0.279	4.706 ± 0.268

TP = translucency parameter, CR = contrast ratio, OP = opalescence parameter, L* = the lightness coordinate, a* = the chromatic coordinate on the red/green axis, b* = the chromatic coordinate on the yellow (positive value)/blue axis, CeZ = Ceramill Zolid fx ML, STM = STML, IPZ = IPS e.max ZirCAD CEREC/in Lab MT Multi, pc = polished-control, gc = glazed-control, c = cervical, m = medium, i = incisal.

**Table 3 jfb-14-00068-t003:** Mean values with SD for L*, a*, b*, TP, CR, OP—after aging.

Variable	Area	CeZ pa	CeZ ga	STM pa	STM ga	IPZ pa	IPZ ga
L*	c	77.688 ± 0.213	81.038 ± 0.217	76.938 ± 0.269	79.075 ± 0.267	77.688 ± 0.218	84.025 ± 0.219
	m	78.463 ± 0.326	81.463 ± 0.278	78.575 ± 0.187	80.513 ± 0.288	78.463 ± 0.348	83.225 ± 0.278
	i	79.863 ± 0.234	83.175 ± 0.219	82.025 ± 0.267	82.925 ± 0.218	79.863 ± 0.289	83.563 ± 0.219
a*	c	−0.025 ± 0.015	0.038 ± 0.005	−0.550 ± 0.005	−0.525 ± 0.028	−0.025 ± 0.008	0.275 ± 0.102
	m	−0.088 ± 0.028	−0.675 ± 0.003	−1.413 ± 0.024	−1.875 ± 0.109	−0.888 ± 0.102	0.100 ± 0.014
	i	−1.088 ± 0.109	−1.013 ± 0.005	−1.988 ± 0.054	−2.663 ± 0.101	−1.088 ± 0.004	−1.538 ± 0.005
b*	c	9.538 ± 0.267	11.000 ± 0.188	27.613 ± 0.185	26.950 ± 0.26	9.538 ± 0.317	16.375 ±0.280
	m	5.125 ± 0.234	5.925 ± 0.207	23.550 ± 0.218	21.700 ± 0.187	5.125 ± 0.218	16.725 ±0.250
	i	1.625 ± 0.154	2.200 ± 0.250	15.425 ± 0.207	12.200 ± 0.267	1.625 ± 0.280	5.525 ± 0.280
TP	c	13.854 ± 0.142	13.766 ± 0.21	13.551 ± 0.132	13.277 ± 0.26	13.749 ± 0.153	13.462 ± 0.187
	m	14.022 ± 0.163	14.007 ± 0.166	13.617 ± 0.188	13.351 ± 0.109	13.636 ± 0.165	13.374 ± 0.188
	i	14.486 ± 0.176	14.325 ± 0.14	14.034 ± 0.174	13.773 ± 0.158	14.135 ± 0.182	13.929 ± 0.128
CR	c	0.693 ± 0.005	0.696 ± 0.003	0.727 ± 0.006	0.732 ± 0.004	0.701 ± 0.003	0.714 ± 0.002
	m	0.692 ± 0.008	0.692 ± 0.004	0.721 ± 0.004	0.722 ± 0.004	0.702 ± 0.004	0.715 ± 0.002
	i	0.676 ± 0.003	0.678 ± 0.005	0.693 ± 0.008	0.704 ± 0.005	0.676 ± 0.005	0.691 ± 0.003
OP	c	5.071 ± 0.269	5.842 ± 0.267	8.184 ± 0.187	7.619 ± 0.126	6.405 ± 0.154	6.314 ± 0.218
	m	4.186 ± 0.250	5.896 ± 0.185	8.170 ± 0.218	7.452 ± 0.234	6.540 ± 0.217	6.383 ± 0.277
	i	3.313 ± 0.207	4.135 ± 0.126	5.852 ± 0.248	6.109 ± 0.236	4.179 ± 0.248	4.865 ± 0.213

TP = translucency parameter, CR = contrast ratio, OP = opalescence parameter, L* = the lightness coordinate, a* = the chromatic coordinate on the red/green axis, b* = the chromatic coordinate on the yellow (positive value)/blue axis, CeZ = Ceramill Zolid fx ML, STM = STML, IPZ = IPS e.max ZirCAD CEREC/in Lab MT Multi, pa = polished-autoclaved, ga = glazed-autoclaved, c = cervical, m = medium, i = incisal.

**Table 4 jfb-14-00068-t004:** Pearson correlation and regression analyses (variables L*, a*, b*) for STM, IPZ (c, m).

Material	L*	a*	b*
STM p, c	very weak, r = 0.196	moderate, r = 0.547	weak, r = 0.235
	r^2^ = 0.038 = 3.84%	r^2^ = 0.299 = 29.9%	r^2^ = 0.055 = 5.5%
	*p* = 0.043	*p* = 0.003	*p* = 0.026
STM p, m	weak, r = −0.341	weak, r = − 0.235	moderate, r = 0.460
	r^2^= 0.116 = 11.6%,	r^2^ = 0.055 = 5.5%,	r^2^ = 0.211 =21.1%,
	*p* = 0.913	*p* = 0.026	*p* = 0.002
IPZ p, c	very strong, r = −0.867	moderate, r = −0.435	very strong, r = 0.800
	r^2^ = 0.751 = 75.1%	r^2^ = 0.189	r^2^ = 0.640 =64%
	*p* < 0.001	*p* < 0.001	*p* < 0.001
IPZ p, m	very weak, r = −0.180	very weak, r = −0.116	very weak, r = 0.068
	r^2^= 0.032 = 3.2%	r^2^ = 0.013 = 1.3%	r^2^ = 0.004 = 0.4%
	*p* = 0.233	*p* = 0.150	*p* = 0.055

L* = the lightness coordinate, a* = the chromatic coordinate on the red/green axis, b* = the chromatic coordinate on the yellow (positive value)/blue axis, r = Pearson coefficient, r^2^ = coefficient of determinations, STM = STML, IPZ = IPS e.max ZirCAD CEREC/in Lab MT Multi, p = polished, c = cervical, m = medium.

**Table 5 jfb-14-00068-t005:** Pearson correlation and regression analyses (variables a*, b*).

Material	a*	b*
CeZ p, c	weak, r = −0.203, r^2^ = 0.041 = 4.1%, *p* = 0.830	moderate, r = 0.567, r^2^ = −0.321 = 32.1%, *p* = 0.001
CeZ p, m	moderate, r = −0.044, r^2^ = 0.002 = 0.2%, *p* = 0.246	weak, r = −0.313, r^2^ = 0.098 = 0.98%, *p* = 0.820
STM p, c	moderate, r = 0.547, r^2^= 0.299 = 29.9%, *p* = 0.003	weak, r = 0.235, r^2^ = 0.055 = 5.5%, *p* = 0.026
STM p, m	weak, r = −0.235, r^2^ = 0.055 = 5.5%, *p* = 0.026	moderate, r = 0.460, r^2^= 0.211 = 21.1%, *p* = 0.002
IPZ p, i	very weak, r = −0.096, r^2^ = 0.009= 0.9%, *p* = 0.147	moderate, r = 0.424, r^2^ = 0.180 = 18%, *p* = 0.053
CeZ g, m	weak, r = 0.020, r^2^ < 0.001 *p* < 0.001	very weak, r = 0.009, r^2^ < 0.001, *p* < 0.001
STM g, m	weak, r = −0.323, r^2^ = 0.104 = 10.4%, *p* = 0.662	strong, r = 623, r^2^ = 0.389 = 38.9%, *p* = 0.031

a* = the chromatic coordinate on the red/green axis, b* = the chromatic coordinate on the yellow (positive value)/blue axis, r = Pearson coefficient, r^2^ = coefficient of determinations, CeZ = Ceramill Zolid fx ML, STM = STML, IPZ = IPS e.max ZirCAD CEREC/in Lab MT Multi, p = polished, g = glazed, c = cervical, m = medium.

**Table 6 jfb-14-00068-t006:** Levels of color changes after aging.

ΔE		CeZ p	CeZ p	STM p	STM g	IPZ p	IPZ g
black	c	1.152	0.587	1.517	0.872	2.239	0.573
m	0.748	0.337	2.564	1.162	2.473	1.15
i	0.188	0.402	2.871	1.132	2.474	0.486
white	c	1.486	0.635	1.644	0.922	2.259	0.598
m	0.938	0.465	2.642	1.381	1.989	1.114
i	0.141	0.289	2.964	1.07	2.141	0.952

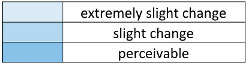
ΔE*—The total color change value. CeZ = Ceramill Zolid fx ML, STM = STML, IPZ = IPS e.max ZirCAD CEREC/in Lab MT Multi, p = polished, g = glazed, m = medium, c = cervical, i = incisal.

## Data Availability

Not applicable.

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
