# Peer review of "The Effect of Polishing, Glazing, and Aging on Optical Characteristics of Multi-Layered Dental Zirconia with Different Degrees of Translucency"

_jfb, 2023, doi:10.3390/jfb14020068_

Round 1

Reviewer 1 Report (New Reviewer)

Title: The Effect of Surface Treatments and Aging on Optical Characteristics of Multi-layered Dental Zirconia with Different Degrees of Translucency

This is a very interesting and informative study regarding how the optical properties of multi-layered zirconia can change after glazing, polishing and aging. Here are some points that should be addressed:

-          In the title it should be better to replace the words “surface treatments” with polishing and glazing” because this term is referred more often to the surface pretreatment of zirconia before cementation and as a result the reader may misunderstand the purpose of the study.

-          I think the tables are too many and may the reader be confused. I suggest removing the tables with the p values to reduce the Results section of the manuscript.

-          Please do not repeat the results of the study in Discussion. This part of the manuscript is to explain the results. Please remove p values and statistics from Discussion. This part should be substantially improved.

-          Please explain in Discussion how artificial aging influences the optical properties. Not the results but the mechanism of action. Moreover, you should explain the differences among the materials and among the surface treatment groups.

-          Please do not repeat the results in Conclusions. Do they have clinical relevance?

Author Response

Review 1

Thank you very much for the review!

- In the title it should be better to replace the words “surface treatments” with polishing and glazing” because this term is referred more often to the surface pretreatment of zirconia before cementation and as a result the reader may misunderstand the purpose of the study.

We agree with this point and we have replaced the words “surface treatments” with “polishing and glazing” in the title.

The Effect of Polishing, Glazing and Aging on Optical Characteristics of Multi-layered Dental Zirconia with Different Degrees of Translucency

-  I think the tables are too many and may the reader be confused. I suggest removing the tables with the p values to reduce the Results section of the manuscript.

We agree with this point and we removed the tables with the p values from the Results section.

 -  Please do not repeat the results of the study in Discussion. This part of the manuscript is to explain the results. Please remove p values and statistics from Discussion. This part should be substantially improved.

We agree with this point and we removed yhe information related to statistics and Results and improved the  Discussion section

-  Please explain in Discussion how artificial aging influences the optical properties. Not the results but the mechanism of action. Moreover, you should explain the differences among the materials and among the surface treatment groups.

We agree with this point and we added additional explanations

ʺThe LTD treatment activate and accelerate the t to m phase transformation, associated with deterioration of the surface exposed to moisture…An increased in translucency, explained by the reduction of light scattering from the boundaries of the cubic phase particles [41], was also observed. However, for the super-high translucency zirconia, the presence of the cubic phase, induce an insignificant hydrothermal degradation with a minor increase in translucencyʺ

ʺAn explanation for the decrease in TP and OP values for polished CeZ, would be that the high roughness of the material negatively affects the appearance of the surface. ʺ

ʺthe glaze layer adhered better and it had a more protective role, even if the surface roughness has increased, nevertheless this material is most affected by aging accompanied by an increase in translucency (more evident on the polished surfaces). ʺ  

 Please do not repeat the results in Conclusions. Do they have clinical relevance?

We agree with this point and we reformulated the conclusions

Reviewer 2 Report (New Reviewer)

Dear authors,

According to my peer review, the following manuscript entitled - The Effect of Surface Treatments and Aging on Optical Characteristics of Multi-layered Dental Zirconia with Different Degrees of Translucency addresses a pertinent topic about prosthetic materials (namely different types of monolithic zirconia) and fall within the scope of Journal of Functional Biomaterials.

After reading your manuscript I found minor aspects that should be corrected or clarified before being accepted for publication.

1)    An updated review/presentation of the topic, different types of monolithic zirconia and their potential optical properties/color changes in the Introduction was provided.

2) Authors should give readers an abbreviation list. After reading the abstract some terms are provided without any description/meaning.

3) Do you really think the information content from line 87 to 94 is absolutely relevant for the Introduction. Or instead could be added to Discussion.

4) The null hypothesis could be better clarified and written;

5) Some aspects of Methods could be also clarified: how many operators preformed the sample preparation?, how many operators read the results? and how can authors justify this sample size?

6)  Authors are invited to attach images of specimen preparation used in this research work.

7) The first sentence of your Results (line 192-193) sounds to be related with Methods.

8) Further limitations of your research should be better discussed. 

kind regards,

Author Response

Review 2

Thank you very much for the review!

  • An updated review/presentation of the topic, different types of monolithic zirconia and their potential optical properties/color changes in the Introduction was provided.

We agree with this point and we restructured and improved the Introduction section

  • Authors should give readers an abbreviation list. After reading the abstract some terms are provided without any description/meaning.

We agree with this point and we added abbreviations in Abstract and Materials and methods sections:

ʺCeramill Zolid fx ML (5mol%) = CeZ, STML (4mol%) = STM, IPS e.maxZirCAD CEREC/in Lab MT Multi (4mol%+5mol%) = IPZʺ

ʺCeramill Zolid fx ML (Amman Girrbach, AG, Koblach, Austria); STML (Katana, Kuraray Noritake Dental, Tokyo, Japan); IPS e.maxZirCAD CEREC/in Lab MT Multi (IvoclarVivadent AG, Schaan, Liechtenstein), marked with the following abbreviations: CeZ, STM, respectively IPZʺ

  • Do you really think the information content from line 87 to 94 is absolutely relevant for the Introduction? Or instead could be added to Discussion.

 We agree with this point and we moved the paragraph in Discussion section

       4) The null hypothesis could be better clarified and written;

            We agree with this point and we reformulated the null hypotheses

ʺ (a) the optical properties of the materials are not influenced by the surface treatments – before or after aging; (b) the translucency and color stability of the zirconia is affected by the artificial aging procedure. ʺ

5) Some aspects of Methods could be also clarified: how many operators preformed the sample preparation? how many operators read the results? and how can authors justify this sample size?

We agree with this point and we added the requested information:

ʺThe samples were performed by a specialized dental technician and the treated surfaces (polished/glazed) were evaluated before and after aging... ʺ

ʺThe measurements were performed by a single operator. ʺ

 ʺThe final dimensions of the specimens, coresponding to the size of the blocks, were 12 mm×14 mm×1 mmʺ

6)  Authors are invited to attach images of specimen preparation used in this research work.

We agree with this point and we have attached an image of the samples

7) The first sentence of your Results (line 192-193) sounds to be related with Methods.

 We agree with this point and we removed this sentence

8) Further limitations of your research should be better discussed.

We agree with this point and we have completed the information

ʺthe limitations of this study would be, the few investigated multi-layered zirconia, only one type of glaze paste was used, the samples had the same uniform thickness, the short exposure time in the autoclave and the effectiveness of the aging method, the lack of certain existing condition (mechanical/chemical stimuli) in oral environment. ʺ

Reviewer 3 Report (New Reviewer)

Thank you for your interesting work. Few comments were attached in the PDF but the main drawback is the Discussion section. Half of it is more suitable in the introduction section and the other half is repetition of results. Very few findings were explained.

Regards

Author Response

Review 3

Thank you very much for the review!

  • [16], [26] Unjustified self citation

We agree with this point and we removed the citation

  • Transfer the chart to linie 121

We agree with this point and we transferred the chart to linie 121

  • Font size

We agree with this point and we corrected the font size

  • The Discussion needs to be more focused. Kindly remove the repetition mentioned in the Introduction section.

We agree with this point and we removed the information that can be found in the Introduction and systemstized the Discussion section.

  • Repetition of the results

We agree with this point and we removed the information that is repeated as well as statistical test

Round 2

Reviewer 3 Report (New Reviewer)

Thank you for addressing the highlighted issues. Kindly check the attached file for the new comments.

Regards

Author Response

Thank you very much for the review!

We agreed with these issues and added the requested information and correction to the text.

This manuscript is a resubmission of an earlier submission. The following is a list of the peer review reports and author responses from that submission.

Round 1

Reviewer 1 Report

Dear Authors,

Thank you very much for submitting the manuscript JFB-2003323 to the Journal of Functional Biomaterials. The research topic interests the readers and the dental materials field.

However, the manuscript needs significant modifications and improvement.

The research design is not appropriate. The variables (dependent and independent) are not stated. The authors described alternative hypotheses - not null hypotheses, as written.

The introduction is very long, and it is recommendable to provide the rationale for the aging method. This method is not so commonly (hydrothermal aging protocol) used in Dentistry compared to thermomechanical cycling. The sample size calculation needs to be described.

The results reflect the issues related to the study design since the statistical analysis is defined along with it. Consequently, the discussion might need revision after the new statistical assessment.

The recommended standard to evaluate color and color changes is the CIEDE 2000. Please justify the choice for the CIELAB. The conditions of the color measurement (measuring geometry, observer, number of repetitions, etc) must be described.

The number of references is excessive, please reduce them.

English proofreading would benefit the fluency of the text.

Author Response

Review 1

Thank you very much for the review!

The research design is not appropriate. The variables (dependent and independent) are not stated. The authors described alternative hypotheses - not null hypotheses, as written.

We agree to this point and we added information related to the variables in this study, we corrected the null hypothesis and changed the research design:

ʺThe purpose of the study was to evaluate the optical properties - the dependent variables TP, CR, OP and color stability (ΔE), among layers with different microstructure, of three types of monolithic multi-layered translucent zirconia, related to surface treatmentʺ

ʺThe null hypotheses established for this study were: (a) polishing and glazing-as surface treatment, does not influence the optical properties of the materials; (b) artificial aging does not affect the translucency and color stability of the zirconia. ʺ

The introduction is very long, and it is recommendable to provide the rationale for the aging method. This method is not so commonly (hydrothermal aging protocol) used in Dentistry compared to thermomechanical cycling.

We agree to this point and we arranged the introduction section and added information about autoclaving as aging method

ʺsteam autoclaves are widely used to achieve accelerated aging of the material [21], according to ISO 13356-2015, dental zirconia is considered admissible for biomedical use when the m-phase content is lower than 25% after 5 hours of autoclave aging procedure (134°C, 2 bars) ʺ

The sample size calculation needs to be described.

We agree to this point and we described the sample size calculation.

ʺUsing descriptive statistics, in the first part-was calculated the dispersion parameters and central tendency. The normal distribution of the data was verified with Shapiro-Wilk test; the non-parametrical and parametrical tests, have been applied for a complete perspective. The two-way ANOVA test was applied for statistical analysis of more than two dependent groups (the areas of a treated surface); the unpaired student T-test, for two different groups (the variables registered on polished and glazed surfaces of a material) and the paired student T-test for two time moments states (to compare the areas of a material - before and after aging). ʺ

The results reflect the issues related to the study design since the statistical analysis is defined along with it. Consequently, the discussion might need revision after the new statistical assessment.

We agree to this point and we arranged the discussion section

The recommended standard to evaluate color and color changes is the CIEDE 2000. Please justify the choice for the CIELAB.

We chose to use CIELAB because:

CIEDE2000 was recently developed and although studies show that it better captures the difference in color, it is more computationally involved, sophisticated and less known to readers...

The conditions of the color measurement (measuring geometry, observer, number of repetitions, etc) must be described.

We agree to this point and we added the information about conditions of the color measurement:

ʺThe instrument was calibrated before each measurement, and then the probe tip was held at 90° to the surface of the sample. The recording was accepted when two identical consecutive readings were obtained on each area (cervical, incisal, medial) in three randomly selected zones of the polished or glazed surfaces. The measurements were performed by a single operator. ʺ

The number of references is excessive, please reduce them.

We agree to this point and we reduced a few, although in another review we were suggested to add new titles and information

English proofreading would benefit the fluency of the text

We agree to this point and we corrected the English mistakes in the manuscript

Reviewer 2 Report

This is an interesting in vitro study of clear novelty and sound design. The  research can open the door for future research in assessing modern ceramics specially translucent zirconia that  increasingly drawing interest of researchers. Please add a few sentences at the end of discussion section to provide  recommendations  for future research specially for the oral environmental conditions and possibility of long term degradation in optical properties.  

Author Response

Review 2:

Thank you very much for the review!

Please add a few sentences at the end of discussion section to provide recommendations for future research specially for the oral environmental conditions and possibility of long term degradation in optical properties.  

We agree with this point and added some informations about the effects of long term degradation on optical properties and recommendation for future research:

ʺ The artificial aging procedures are used to clarify and predict the long-term degradation on the mechanical and optical properties of zirconia. A correlation between translucency, mechanical properties and the thickness of the material, it was observed, also the influence of the composition and grain size on the surface treatments [11,25]. In order to obtain a natural appearance, different dopants and stabilizers are used, that affect the optical properties during the aging process. According to these findings, future studies are needed to investigate the behaviour of the dopants over time; the appearance and the changes that occurred depending on the thickness of the material. Because it has an almost similar hardness, but a lower translucency than lithium disilicate glass-ceramics or enamel, a novel alternative as zirconia-reinforced lithium silicate glass-ceramic can be taken in consideration.ʺ

Reviewer 3 Report

Dear Authors,

this paper summarize a relevant amount of experimental work, and deems to me suitable for publication in this journal. Nertheless, it contains several typing errors, and errors in formatting of the text that you should revise before publication. 

In addition, I suggest

- to introduce in the sample characterization the measure of the size of the grains and of its distribution, taking into account the relevance of this parameter on the optical properties;

- to add the measure of the content in cubic (translucent) phase in each one of the materials tested. This because the LTD process affects only the tetragonal phase of zirconia, leaving the cubic one unaffected.

Author Response

Review 3:

Thank you very much for the review!

it contains several typing errors, and errors in formatting of the text that you should revise before publication. 

We agree to this point and we corrected the errors in the text

- to introduce in the sample characterization the measure of the size of the grains and of its distribution, taking into account the relevance of this parameter on the optical properties;

-to add the measure of the content in cubic (translucent) phase in each one of the materials tested

We agree to this point and we added a column with the size of the granules in the presentation table of the materials. Some values are not disclosed by the manufacturers and we have centralized the data of some studies about these materials.

ʺ5Y-TZP (the super-high translucent) - partially stabilised zirconia with 5 mol% Y₂O₃ and more than 50% cubic phase, respectively 4Y-TZP (the super-translucent) - partially stabilised zirconia with 4 mol% and more than 25% cubic phase ʺ

ʺ Summarizing the reports of several studies on different types of zirconia, the super translucent one presents particles with a size of 0.5-1 µm and a cubic phase content of 25-53%, and the high translucent zirconia may content grains larger than 1 µm-differently distributed and an amount of 60-71% cubic phase, making them highly transparent but more susceptible to fracture ʺ

Reviewer 4 Report

Dear Authors,

generally, the write-up of the paper is good. Here follow some suggestions to improve the manuscript.

Please review English with the help of a proofreader.

  1. Lines 40-43:

After this paragraph, the authors shall explain the reader also the negative sides of this material.

Please add limitations of SMTL zirconia such as “lower marginal adaptation after cyclic fatigue” (you could cite: https://doi.org/10.1111/jerd.12837 ) and “lower optical properties” (you could cite: https://doi.org/10.1016/j.heliyon.2021.e08151 ) in respect to lithium silicate or disilicate.

  1. This is just a suggestion: generally, the hypothesis is considered not to have differences among groups. But again, this is not mandatory.

  1. Line114:

The authors wrote: “glazing or polishing-as surface treatment, it influences the optical properties of the materials”

Please modify in: “glazing or polishing-as surface treatment influences the optical properties of the materials”

  1. Line 140: please state if the measure check was performed by a single operator or not.

  1. Line 175: please rephrase in: “ΔE* formula is used to evaluate color difference: before and after aging.”
  2. Remove bold formatting
  3. Line 450: remove bold formatting
  4. Please add a limitation of the Spectrophotometer used:

Easyshade (Vita Zahnfabrik, Bad Säckingen, Germany)  is a clinical device (only working in "tooth mode") and it is generally not recommended for in-vitro testing. (you could cite: https://doi.org/10.1016/j.jdent.2022.104223)

Author Response

Review 4:

Thank you very much for the review!

Please review English with the help of a proofreader.

We agree to this point and we corrected the English errors in the manuscript.

Lines 40-43:

After this paragraph, the authors shall explain the reader also the negative sides of this material.

We agree to this point and we added the information:

ʺFor 5Y-TZP, the amount of tetragonal phase decreased leading to a higher translucency and a lower ability of t to m phase transformation, with reduction in toughness. The main phase amount, implicitly the microstructure, may influence some clinical features of zirconia: translucency, strength, resistance to degradation (aging) [5]. ʺ

Please add limitations of SMTL zirconia such as “lower marginal adaptation after cyclic fatigue” (you could cite: https://doi.org/10.1111/jerd.12837 ) and “lower optical properties” (you could cite: https://doi.org/10.1016/j.heliyon.2021.e08151 ) in respect to lithium silicate or disilicate.

We agree to this point and we added the information and we cited the two sources:

ʺ Because zirconia, in particular 4Y-TZP, has an almost similar hardness [60], but a lower translucency [61], than lithium disilicate glass-ceramics, ʺ

 the hypothesis is considered not to have differences among groups

We agree to this point and we reformulated the hypothesis

  1. Line114:

The authors wrote: “glazing or polishing-as surface treatment, it influences the optical properties of the materials”

Please modify in: “glazing or polishing-as surface treatment influences the optical properties of the materials”

We agree to this point and we corrected the information:

ʺ glazing or polishing-as surface treatment, influences the optical properties of the materials ʺ

please state if the measure check was performed by a single operator or not.

We agree to this point and we added the information:

ʺ The measurements were performed by a single operator. ʺ

please rephrase in: “ΔE* formula is used to evaluate color difference: before and after aging.”

We agree to this point and we corrected the information:

ʺ ΔE*- The total color change value, signify the color difference among two stages, was achieved using the formula: ʺ

 Remove bold formatting

We agree to this point and we removed the bold formatting

Please add a limitation of the Spectrophotometer used:

Easyshade (Vita Zahnfabrik, Bad Säckingen, Germany) is a clinical device (only working in "tooth mode") and it is generally not recommended for in-vitro testing. (you could cite: https://doi.org/10.1016/j.jdent.2022.104223

We agree to this point and we added the information and cited the recommended source:

ʺ spectrophotometer Easyshade IV (Vita Zahnfabrik, Bad Säckingen, Germany), is a clinical device that working only in "tooth mode" [31]. ʺ

Round 2

Reviewer 1 Report

The issues raised in the review were not addressed.

Author Response

(The authors gave the same response as above.)

Reviewer 3 Report

Dear Author,

I am afraid that you missed the key point in my former comments. You have not to rely on manufacturer data (only) for key parameter of your study (in this case grain size and phase content). This is because you do not know the confidence interval that each manufacturer accepts nor if the values released are average, minimum or else.

Then, you have to make your own measurements of data which are the keystone of your conclusions, i.e. grain size and cubic phase content in the materials under study.

In addition, the quantification of phase shift from tetragonal to monoclinic occurring during aging (LTD) - a behavior that you cite in the conclusion of the manuscript - is mandatory, because it may justify some of the changes in the optical properties observed. 

Would you please add the description of the Methods used for the measure of the parameters in above.

Author Response

(The authors gave the same response as above.)
